# The Prognostic Value of Neutrophil-to-Lymphocyte Ratio in Patients with Metastatic Renal Cell Carcinoma

**Andreea Ioana Parosanu** [1,2]**, Cristina Florina Pirlog** [1,2]**, Cristina Orlov Slavu** [1,2]**, Ioana Miruna Stanciu** [1,2]**,
Horia-Teodor Cotan** [1,2]**, Radu Constantin Vrabie** [1,2]**, Ana-Maria Popa** [1,2]**, Mihaela Olaru** [1,2]**, Cristian Iaciu** [1,2]**,
Lucian Ioan Bratu** [3,4]**, Ionut Florian Baicoianu** [3,4]**, Oana Moldoveanu** [3,4]**, Catalin Baston** [3,4,*]** and Cornelia Nițipir** [1,2]

[1]  Department of Medical Oncology, Elias Emergency University Hospital, 011461 Bucharest, Romania
[2]  Department of Oncology, Faculty of Medicine, "Carol Davila" University of Medicine and Pharmacy, 050474 Bucharest, Romania
[3]  Department of Urology, Fundeni Clinical Institute, 022328 Bucharest, Romania
[4]  Department of Urology, Faculty of Medicine, "Carol Davila" University of Medicine and Pharmacy, 050474 Bucharest, Romania
[*]  Correspondence: catalin.baston@umfcd.ro; Tel.: +4-0765210001

**Abstract:** Background: Metastatic renal cell carcinoma (mRCC) is an aggressive cancer characterised by an increased recurrence rate and an inadequate response to treatment. This study aimed to investigate the importance of the neutrophil-to-lymphocyte ratio (NLR) as a prognostic marker for long-term survival in patients with mRCC. Methods: We retrospectively analysed data from 74 patients with mRCC treated at our medical centre with tyrosine kinase inhibitors (TKIs) and immune checkpoint inhibitors (ICIs). We evaluated the predictive value of NLR for overall survival (OS) in these patients. Results: The median OS was 5.1 months in the higher NLR group ($\geq$3) and 13.3 months in the lower NLR group (<3) ($p < 0.0001$). There was no significant difference in the OS between the TKI and ICI therapies in the low NLR group (12.9 vs. 13.6 months, $p = 0.411$) or in the high NLR group (4.7 vs. 5.5 months, $p = 0.32$). Both univariate and multivariate analyses revealed that a higher NLR was an independent prognostic factor of long-term survival in patients with mRCC treated with first-line therapy. Conclusions: This retrospective study showed that adding NLR to other Memorial Sloan Kettering Cancer Center (MSKCC) and International Metastatic Renal Cell Carcinoma Database Consortium (IMDC) variables might improve the prognostic and predictive power of these models.

**Keywords:** NLR; MSKCC; IMDC; metastatic; renal cell carcinoma

## 1. Introduction

The most common histological type of renal cell carcinoma (RCC) is clear cell carcinoma (ccRCC), which accounts for approximately 85% of all cases. It is the sixth most common type of cancer in men and the ninth most common type in women [1,2].

Renal cell carcinoma is a well-vascularized and immunogenic tumour characterized by a massive infiltration of various immune cells. Consequently, the current therapeutic approaches include anti-angiogenic agents, cancer immunotherapy, or both [3,4].

An elevated neutrophil-to-lymphocyte ratio (NLR) can reflect both the presence of neutrophilia and lymphopenia and may suggest impaired cell-mediated immunity in patients with cancer. Therefore, NLR is considered a robust prognostic biomarker in certain tumours, including digestive or genitourinary cancers [5–7].

Hence, can we incorporate the NLR, which is easily calculated using complete blood cell counts and widely measured in daily clinical practice?

Even reporting the clinical experience of a small number of patients may aid in the identification of potential additional biomarkers for predicting survival and enhancing patient management.

## 2. Materials and Methods

We retrospectively analysed 74 eligible patients with metastatic renal clear cell carcinoma treated at our department of medical oncology at the Elias Emergency University Clinic Hospital, Bucharest, Romania, from the 1 January 2020 to the 31 October 2022.

The selection criteria were as follows: a histologic diagnosis of metastatic or locally advanced unresectable RCC, clear cell histology and aged over 18 years. Informed consent was obtained from all subjects involved in the study. The study was conducted in accordance with the Declaration of Helsinki, and approved by the Institutional Review Board of Elias Emergency University Clinic Hospital (no. 7170/12 January 2023)

All the patients were deemed eligible for first-line therapy with tyrosine kinase inhibitors or immunotherapy, depending on the risk assessed with the IMDC and MSKCC prognostic models.

The study population was stratified into NLR low (<3) or NLR high (>3) according to a cut-off point value established at three. We performed the receiver-operating characteristic (ROC) curve analysis by calculating the area under the curve (AUC) to determine the specific cut-off values of NLR. Univariate and multivariate analyses were used to evaluate factors influencing the response to first-line therapy. The parameters analysed were age at diagnosis, gender, tumoral stage, histology, metastatic sites, various serum variables, and the neutrophil-to-lymphocyte ratio (NRL). The data were analysed using the Kaplan–Meier method and log-rank tests. Statistical significance was established when $p < 0.05$.

## 3. Results

### 3.1. Patient Characteristics

All the patients had a clear cell histology. The clinicopathological features of the 74 patients are listed in Table 1.

**Table 1.** The clinical–pathological characteristics.

| Characteristics | *N* (%) |
|---|---|
| All patients | 74 |
| Age (years) | 62.8 (range 43–88) |
| Gender<br>Male<br>Female | <br>52 (70.3%)<br>22 (29.7%) |
| Surgical treatment<br>Radical nephrectomy<br>Tumour biopsy<br>Partial nephrectomy | <br>48 (64.8%)<br>11 (14.8%)<br>15(20.2%) |
| The main sites of metastasis<br>Lungs<br>Distant lymph nodes<br>Liver<br>Bones | <br>17 (23%)<br>10 (13.5%)<br>29 (39.2%)<br>21 (28.4%) |
| Fuhrman grade<br>2<br>3<br>4 | <br>35 (47.3%)<br>33 (44.5%)<br>6 (8.1%) |
| Karnofsky Performance Status<br><80%<br>≥80% | <br>17 (23%)<br>57 (77.0%) |

**Table 1.** *Cont.*

| Characteristics | *N* (%) |
|---|---|
| Time since diagnosis to treatment | |
| <12 months | 52 (70.3%) |
| ≥12 months | 22 (29.7%) |
| Haemoglobin | |
| <LLN | 41 (55.4%) |
| ≥LLN | 33 (44.6%) |
| LHD | |
| ≥1.5× ULN | 12 (16.2%) |
| <1.5× ULN | 62 (83.8%) |
| Serum-corrected calcium | |
| ≥ULN | 16 (21.6%) |
| <ULN | 58 (78.4%) |
| Platelet count | |
| ≥ULN | 24 (32.4%) |
| <ULN | 50 (67.6%) |
| Neutrophil count | |
| ≥ULN | 11 (14.9%) |
| <ULN | 62 (83.8%) |
| NLR | |
| Median (range) | $3.34 \pm 3.06$ (1–22) |
| ≥3 | 33 (44.5%) |
| <3 | 41 (55.4%) |
| IMDC score | |
| Favourable | 5 (6.8%) |
| Intermediate | 38 (51.4%) |
| Poor | 31 (41.9%) |
| MSKCC score | |
| Low risk | 8 (10.8%) |
| Intermediate risk | 49 (66.2%) |
| High risk | 17 (23.0%) |

LLN (lower limit of normal), ULN (upper limit of normal).

### 3.2. The Relationship between Clinicopathological Parameters and Survival

According to the univariate analysis, poor cancer-specific survival had significant relationships with a Karnofsky score of <80% (HR 11.60, 95%; $p < 0.001$), late treatment initiation (over 12 months) (HR 1.04 95% CI $p = 0.009$), haemoglobin < LLN (HR 5.52 CI 95% $p = 0.002$), LHD over 1.5 times ULN (HR 3.31, 95% CI $p = 0.002$), an NLR of ≥3 (HR 10.31, 95% CI $p < 0.001$) and high IMDC and MSKCC scores (HR 6.11, 95% CI, $p < 0.001$) (Supplementary Table S1, Table 2).

On multivariate analysis, only a Karnofsky score of <80% (HR 16.008, 95% CI $p = 0.009$), a time from diagnosis to the start of systemic treatment of >12 months (HR 10.819, 95% CI $p = 0.0011$) and an NLR of ≥ 3 (HR 4.650, 95% CI $p = 0.006$) were significantly and independently associated with inferior overall survival (Table 3, Figure 1).

**Table 2.** Univariate analysis.

| Variable | *p*-Value | Hazard Ratio | 95% Confidence Interval |
|---|---|---|---|
| Karnofsky Performance Status <80% | 0.001 | 2.42 | 1.84–2.76 |
| Time since diagnosis to treatment <12 months | 0.009 | 10.819 | 1.718–68.135 |
| Haemoglobin < LLN | 0.002 | 1.904 | 0.653–5.552 |
| LHD > 1.5× ULN | 0.002 | 1.924 | 0.661–5.597 |
| NLR ≥ 3 | 0.001 | 1.55 | 1.23–1.91 |
| High IMDC and MSKCC scores | 0.001 | 3.30 | 2.22–4.89 |

Age ($p$ = 0.88), gender ($p$ = 0.355), Fuhrman grade ($p$ = 0.085), calcium higher than the upper limit of normal ($p$ = 0.595), platelets and neutrophil counts higher than the upper limit of normal ($p$ = 0.075, and $p$ = 0.102) were not found to be statistically significant in predicting survival.

**Table 3.** Multivariate analysis.

| Variable | *p*-Value | Hazard Ratio | 95% Confidence Interval |
|---|---|---|---|
| Karnofsky Performance Status <80% | 0.009 | 16.008 | 1.989–128.86 |
| Time since diagnosis to treatment <12 months | 0.0011 | 10.819 | 1.718–68.135 |
| NLR ≥ 3 | 0.006 | 4.650 | 1.562–13.840 |

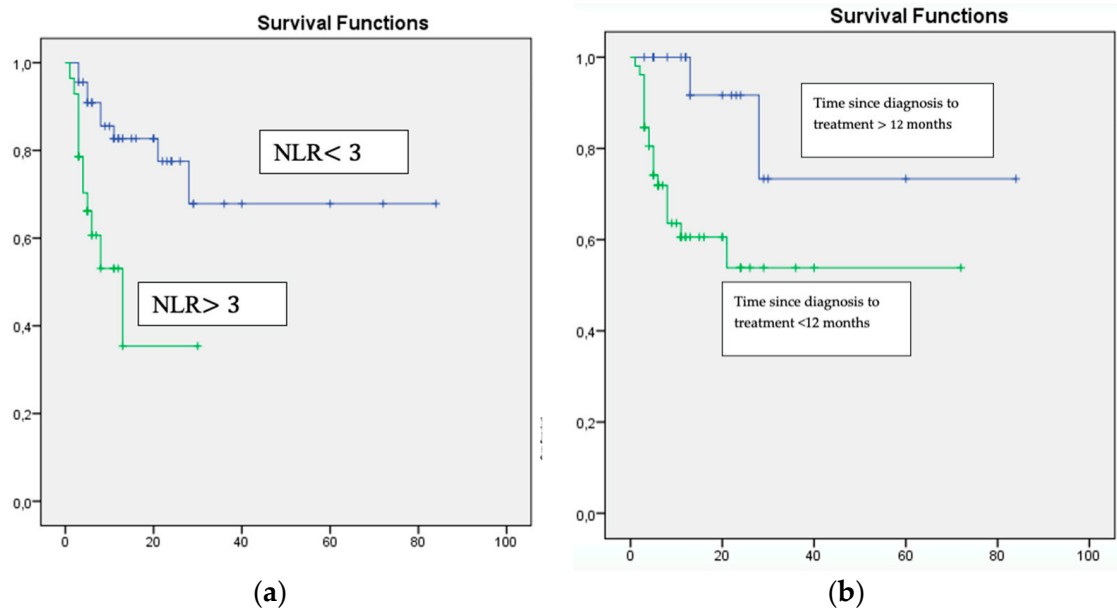

**(a)**      **(b)**

**Figure 1.** *Cont.*

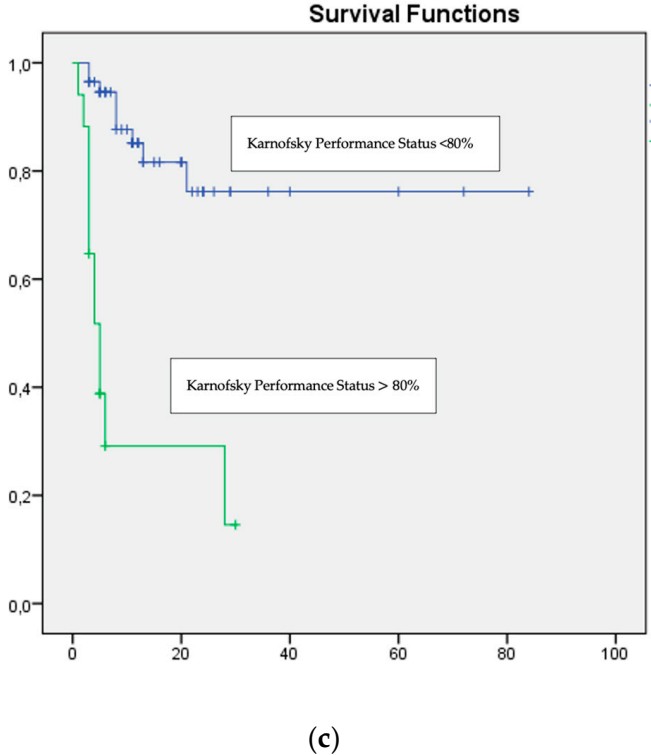

(**c**)

**Figure 1.** Multivariate analysis: (**a**) NLR $\geq$ 3, (**b**) Time since diagnosis to treatment <12 months and (**c**) Karnofsky Performance Status <80%, were associated with unfavourable survival.

Systemic therapy was administered to all the patients after the diagnosis of mRCC. TKIs were used most frequently (*n* = 50, 67.5%). Immunotherapy with Nivolumab and Ipilimumab was used in 24 (32.4 %) patients.

The median follow-up was conducted for 15.3 (range, 4.3–22.6) months. The overall survival for all patients was a median of 12.7 months. In the entire cohort, the median overall survival for patients with an NLR of $\leq$3 was 13.3 months vs. 5.1 months for those with an NLR of >3.

There was no significant difference in the OS between the TKI and ICI therapies in the low NLR group (12.9 months vs. 13.6 months, *p* = 0.411) or in the high NLR group (4.7 months vs. 5.5 months, *p* = 0.32).

## 4. Discussion

Renal cell carcinoma represents 2.4% of all cancer diagnoses, and its incidence has increased globally over the last two decades [8].

Surgery can be a curative procedure for a minority of patients who present with early-stage disease. However, for advanced and metastatic stages, systemic therapy is essential. Renal cell carcinoma is a highly immunogenic and chemotherapy-resistant tumour [9]. Currently, anti-angiogenic drugs and immune checkpoint inhibitors have been established as the new standard of care for patients with mRCC [10,11]. Immuno-oncology-based doublet combinations have a highly significant effect on patients with an intermediate or poor prognosis [12]. Therefore, combination therapy is only the best choice for some patients. However, monotherapy with TKI may be an appropriate treatment option for favourable-risk patients to prevent the potential toxicities associated with ICI therapy [13].

Although most factors that affect the prognosis are related to the tumour pathology and the patient's clinical and biological characteristics, the potential outcome for each patient remains uncertain.

IMDC and MSKCC criteria are already broadly adopted to estimate RCC patient prognosis. However, can we improve these scores?

Studying the role of cellular inflammatory markers in the interaction between immune response and cancer is challenging. The neutrophil-to-lymphocyte ratio (NLR) reflects a dynamic balance between innate and adaptive immune activity. Therefore, a high NLR suggests chronic inflammation and immune distress [14,15].

Today, NLR is widely reported as a reliable and readily available prognostic marker in various solid cancers, but with no widely accepted cut-off point. Normal NLR values are between one and two. Elevated values, defined as an NLR of $\geq 3$, are regarded as pathological [16,17].

Recent published studies of patients with solid metastatic tumours have shown (using multivariate Cox regressions and time-dependent sensitivity analysis) that the optimal NLR cut-off value varies from 2.5 to 5 [18–20]. Therefore, our cut-off point of three was based on a previous analysis with similar findings [21]. For example, an extensive systematic review and meta-analysis investigated the association between NLR, disease-free progression and overall survival in 18 studies with 2735 patients selected. The results indicated that an elevated pre-treatment NLR of $\geq 3$ was significantly associated with poorer OS and DFS (HR = 2.31, 95% $p < 0.001$) [22].

According to our study, a high baseline NLR ($\geq 3$) was correlated with a worse OS (13.3 months vs. 5.1 months, $p = 0.001$) with no considerable differences between patients treated with TKIs or ICIs (4.7 months vs. 5.5 months, $p = 0.32$).

Our findings agree with those of A. K. A Lalani from Dana–Farber Cancer Institute, who reported a significantly longer survival for patients treated with PD-1/PD-L1 immunotherapy and a low NLR at baseline. In addition, a maintained low NLR after six weeks of treatment further improved outcomes [23]. Similarly, a large systematic review by Chen X also demonstrated that a high NLR at baseline or pre-therapy was significantly associated with a worse overall survival (HR, 2.23; 95% CI, 1.84–2.70; $p < 0.001$) in patients with mRCC treated with ICIs [24].

Several studies evaluating the NLR as a personalized outcome prediction tool in patients treated with TKIs have clearly established an increased NLR value as a negative prognostic factor [25–27]. For instance, A.J. Templeton confirmed in a retrospective analysis of 5549 subjects with mRCC that a higher NLR at baseline was associated with an adverse OS and PFS. In this context, an increase in NLR after six weeks of therapy reassured that the therapy was associated with a good clinical response and better survival [28].

These data support our study's conclusion about the significative predictive value of NLR in patients with RCC receiving both immunotherapy and TKIs.

Despite the rapidly growing body of literature on NLR, the mechanism underlying the association of this marker of inflammation remains poorly understood. Our results encourage the routine monitoring of NLR to predict recurrence, progress and survival outcomes in patients with RCC.

This study has several limitations. First, it is a retrospective analysis of a small number of patients who received different first-line regimens, including TKI or ICIs, according to the approved therapies available in Romania. Moreover, the data were collected during the COVID-19 pandemic, which caused treatment modifications and immensely disrupted the therapies' acceptability and availability.

## 5. Conclusions

A survival prognosis is essential but very challenging. This study confirms that a high pre-treatment neutrophil-to-lymphocyte ratio of $\geq 3$ predicts an unfavourable outcome in patients with advanced RCC treated with first-line ICIs or TKIs. In addition, in our univariate and multivariate models for OS, a poor performance status and $\geq$ one-year interval between diagnosis and treatment initiation were also associated with inferior outcomes. Therefore, NLR may be considered an additional variable that improves the prognostic prediction of the IMDC and MSKCC models. However, a more extensive prospective study is needed to validate these results.

**Supplementary Materials:** The following supporting information can be downloaded at: https://www.mdpi.com/article/10.3390/curroncol30020187/s1, Table S1: Statistical analysis of potential prognostic markers.

**Author Contributions:** Conceptualization, A.I.P.; Methodology, C.B. and C.N.; Formal analysis, A.I.P.; Investigation, A.I.P.; Writing—original draft, A.I.P.; Writing—review and editing, C.O.S.; Supervision, C.F.P., I.M.S., H.-T.C., R.C.V., A.-M.P., M.O., C.I., L.I.B., I.F.B. and O.M. All authors have read and agreed to the published version of the manuscript.

**Funding:** This research received no external funding.

**Institutional Review Board Statement:** The study was conducted in accordance with the Declaration of Helsinki, and approved by the Institutional Review Board of Elias Emergency University Clinic Hospital (no. 7170/12 January 2023).

**Informed Consent Statement:** Informed consent was obtained from all subjects involved in the study. Written informed consent has been obtained from the patients to publish this paper.

**Data Availability Statement:** Data supporting this study are included within the article and supporting materials.

**Conflicts of Interest:** The authors declare no conflict of interest.

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
