# Peer review of "The Prognostic Value of Neutrophil-to-Lymphocyte Ratio in Patients with Metastatic Renal Cell Carcinoma"

_curroncol, doi:10.3390/curroncol30020187_

Round 1
Reviewer 1 Report
Dear Authors,
This manuscript addresses an issue that is very important and relevant to the medical community. I applaud your recognition of the limitations of the study.
There are minor issues that can be improved in this manuscript which will enrich the manuscript.
The following are my comments describing these issues.
INTRODUCTION
The Authors should deepen these aspects in the introduction:
Line 34-37.
It is interesting that more information should be included:
Point 1. The most common subtype of renal cell carcinoma (RCC) is clear cell carcinoma (ccRCC), which accounts for ~85% of all cases (https://doi.org/10.3322/caac.21442). It is the sixth most common type of cancer in men and the ninth most common type in women (doi: 10.3322/caac.21654).
In this way, including the information above supports patients selected for the trial who have ccRCC.
Line 48. “Current first-line treatment decisions for advanced or metastatic RCC depend on risk 45 stratification models, including the Memorial Sloan Kettering Cancer Center (MSKCC) 46 and the International Metastatic Renal Cell Carcinoma Database Consortium (IMDC) 47 models [7]. Interestingly, neutrophil count is part of the IMDC prognostic model score”.
Point. This paragraph implies that this criterion is not important for MSKCC, and MSKCC is also working on this question (https://doi.org/10.1038/s41467-021-20935-9). So, it would be interesting to change the approach of that subsection.
MATERIALS AND METHODS
Line 56. “We retrospectively analysed 74 eligible patients”
Point 1. Where did the patients come from? Include the hospital, the city, and the country.
Point 2. Indicate if the consent form was submitted to all study participants.
Point 3. This trial was approved by (?). Insert name of ethics committee, approval number...
DISCUSSION
Line 125. “Renal cell carcinoma represents only 2% of all cancer diagnoses, but its incidence has increased globally over the last two decades [8]”.
Capitanio U, Bensalah K, Bex A, Boorjian SA, Bray F, Coleman J, Gore JL, Sun M, Wood C, Russo P. Epidemiology of Renal Cell 215 Carcinoma. Eur Urol. 2019 Jan;75(1):74-84. doi: 10.1016/j.eururo.2018.08.036. Epub 2018 Sep 19. PMID: 30243799; PMCID: 216 PMC8397918
Point 1. This epidemiological data in the introduction is more appropriate (Line 34-37).
Point 2. That information can be refined “Renal cell carcinoma represents 2.4% of all cancer diagnoses, and its incidence”. Furthermore, there is a timelier reference (World Health Organization (WHO). Global Health Estimates 2020: Deaths by Cause, Age, Sex, by Country and by Region, 2000–2019; WHO: Geneva, Switzerland, 2020).
Last question: was there any funding for this study?
Kind regards.
Author Response
Dear Reviewer,
We are grateful to the reviewer for the careful evaluation of our manuscript and for the constructive suggestions; all of them helped us to further improve the quality of our manuscript. We appreciate the thorough work of the reviewer and all the comments. We believe that we could fix his suggestions.
Please see the attachment. Thank you!
Kind regards,
Andreea Parosanu

Reviewer 2 Report
This study was reported the utility of NLR as a predictive factor for mRCC. The reviewer would like to suggest some critiques to make this paper as follows.
Major revision
1. The authors should revise the Title. “utility” is wrong.
2. Is the p-value stated correctly? p=0.01? p = 0.01?
3. The other parts of this report refer to renal cell carcinoma, but only line 38 refers to Kidney carcinoma. The authors should use “renal cell carcinoma.”
4. On line 69, the authors need to describe how they determined the cutoff values for the NLR. The reviewer thinks that the AUC should be used to determine the cutoff value.
5. On line 73, “NRL” is wrong.
6. Lines 78 through 89 need to be deleted or significantly modified since they are already in the table.
7. Currently, combination therapy with ICI and ICI or TKI is the mainstream therapy, but the clinical significance of the comparison of monotherapy should be clearly stated in the Discussion.
Author Response

(The authors gave the same response as above.)

Round 2
Reviewer 2 Report
The authors revised the paper in accordance with the reviewers’ comments. The reviewer believes that this paper will provide useful information for readers.